# Repurposing Niclosamide for Targeting Pancreatic Cancer by Inhibiting Hh/Gli Non-Canonical Axis of Gsk3β

**DOI:** 10.3390/cancers13133105

**Published:** 2021-06-22

**Authors:** Jyoti B. Kaushal, Rakesh Bhatia, Ranjana K. Kanchan, Pratima Raut, Surya Mallapragada, Quan P. Ly, Surinder K. Batra, Satyanarayana Rachagani

**Affiliations:** 1Department of Biochemistry and Molecular Biology, University of Nebraska Medical Center, Omaha, NE 68198, USA; jyoti.kaushal@unmc.edu (J.B.K.); rocky.bhatia@unmc.edu (R.B.); ranjana.kanchan@unmc.edu (R.K.K.); pratima.raut@unmc.edu (P.R.); sbatra@unmc.edu (S.K.B.); 2Department of Chemical and Biological Engineering, Nanovaccine Institute, Iowa State University, Ames, IA 50011, USA; suryakm@iastate.edu; 3Department of Surgical Oncology, University of Nebraska Medical Center, Omaha, NE 68198, USA; qly@wnmc.edu; 4Fred & Pamela Buffet Cancer Center, Eppley Institute for Research in Cancer and Allied Diseases, University of Nebraska Medical Center, Omaha, NE 68198, USA

**Keywords:** niclosamide, pancreatic cancer, apoptosis, autophagy, Hh signaling, Gsk3β

## Abstract

**Simple Summary:**

The current obstacles for discovering new drugs for cancer therapy have necessitated the development of the alternative strategy of drug repurposing, the identification of new uses for approved or investigational drugs for new therapeutic purposes. Niclosamide (Nic) is a Food and Drug Administration (FDA)-approved anti-helminthic drug, reported to have anti-cancer effects, and is being assessed in various clinical trials. In the current study, we assessed the therapeutic efficacy of Nic on pancreatic cancer (PC) in vitro. Our results revealed mitochondrial stress and mTORC1-dependent autophagy as the predominant players of Nic-induced PC cell death. This study provided a novel mechanistic insight for anti-cancer efficacy of Nic by increasing p-Gsk3β that modulates molecular signaling(s), including inhibition of hedgehog (Hh) signaling-mediated cellular proliferation and increased apoptosis through mTORC1-dependent autophagy may prove helpful for the development of novel PC therapies.

**Abstract:**

Niclosamide (Nic), an FDA-approved anthelmintic drug, is reported to have anti-cancer efficacy and is being assessed in clinical trials for various solid tumors. Based on its ability to target multiple signaling pathways, in the present study, we evaluated the therapeutic efficacy of Nic on pancreatic cancer (PC) in vitro. We observed an anti-cancerous effect of this drug as shown by the G0/G1 phase cell cycle arrest, inhibition of PC cell viability, colony formation, and migration. Our results revealed the involvement of mitochondrial stress and mTORC1-dependent autophagy as the predominant players of Nic-induced PC cell death. Significant reduction of Nic-induced reactive oxygen species (ROS) and cell death in the presence of a selective autophagy inhibitor spautin-1 demonstrated autophagy as a major contributor to Nic-mediated cell death. Mechanistically, Nic inhibited the interaction between BCL2 and Beclin-1 that supported the crosstalk of autophagy and apoptosis. Further, Nic treatment resulted in Gsk3β inactivation by phosphorylating its Ser-9 residue leading to upregulation of Sufu and Gli3, thereby negatively impacting hedgehog signaling and cell survival. Nic induced autophagic cell death, and p-Gsk3b mediated Sufu/Gli3 cascade was further confirmed by Gsk3β activator, LY-294002, by rescuing inactivation of Hh signaling upon Nic treatment. These results suggested the involvement of a non-canonical mechanism of Hh signaling, where p-Gsk3β acts as a negative regulator of Hh/Gli1 cascade and a positive regulator of autophagy-mediated cell death. Overall, this study established the therapeutic efficacy of Nic for PC by targeting p-Gsk3β mediated non-canonical Hh signaling and promoting mTORC1-dependent autophagy and cell death.

## 1. Introduction

Pancreatic cancer (PC) is a lethal disease with an extremely poor prognosis and a five-year survival rate of less than 8% [1]. Risk factors associated with PC progression and development include family history, smoking, obesity, sudden onset of diabetes, and chronic pancreatitis [2]. PC is projected to become the second leading cause of cancer-related deaths by 2030 [3,4]. It remains a challenging disease due to its aggressive nature, late clinical presentation, early metastasis to local and distant organs, and inherent resistance to current therapies [5,6]. The standard management relies on cytotoxic chemotherapy, primarily fluorinated pyrimidine antimetabolites (gemcitabine, fluorouracil), topoisomerase I inhibitor (irinotecan), DNA crosslinking agents (oxaliplatin, cisplatin), tubulin inhibitors (paclitaxel, nab-paclitaxel) as well as other regimens [7,8,9]. However, these therapeutic options showed limited therapeutic response and outcome in a subset of PC patients and often resulted in high toxicity [8]. Therefore, there is an inordinate requirement for novel anti-cancer agents that provide therapeutic benefit with minimal toxicity to combat lethal metastatic PC. The current obstacles for discovering new drugs via the traditional approach imposed the alternative strategy of drug repurposing. The development of old drugs for new therapeutic purposes ultimately promotes the holistic productivity of drug discovery [9,10]. This method is more economical, faster, good safety profiles, and a considerably higher probability of success at the preclinical and clinical level than canonical/traditional drug development strategy [11].

In this regard, numerous studies were conducted to screen the role of the non-anticancer drug as an anti-cancer drug, out of which anthelmintic drugs including benzimidazole family members (albendazole, mebendazole, and flubendazole), salicylanilides (niclosamide), and cyanine dye derivatives (pyrvinium) are being considered for repurposing in cancer therapy [12,13]. Niclosamide (Nic), a salicylanilide derivative, is a Food and Drug Administration (FDA)-approved chewable tablet consumed orally against schistosomiasis and tapeworm infections in millions of people worldwide over 50 years [13]. The underlying model of Nic action is inhibition of glucose uptake, oxidative phosphorylation, anaerobic metabolism, and stimulation of adenosine triphosphatase activity in the mitochondria of target worms [14]. However, recent studies have shown that Nic has a broad range of clinical applications to treat other diseases and is identified as a potential anti-cancer agent [15]. The anti-cancer activity of Nic is associated with the inhibitory effects on multiple intracellular signaling pathways such as Wnt/β-catenin signaling (adrenocortical carcinoma, ovarian cancer, glioma, colorectal, breast, head and neck, and renal cell carcinoma) [16,17,18,19,20,21,22], JAK/STAT pathway (lung cancer and colon cancer) [23,24] and mTORC1, Notch, NFκB signaling pathways (cervical, colon, glioma, and leukemia) [25,26,27]. Recently, Nic was identified as a repositioned therapeutic agent for PDAC, probably targeting ArfGAP with SH3 domain, ankyrin repeat, and PH domain (ASAP2), a member of the ArfGAP family [28]. The aberrant activation of the molecular signaling pathways such as Wnt/β-catenin, Hh/Gli1, mTORC1 is a common theme across several cancer types, including PC that facilitates metastasis and disease recurrence [29]. Thus, targeting these multiple potential molecular targets is a promising approach to combat lethal PC.

In the current study, our main goal was to determine the anti-cancerous activity of Nic in PC and to delineate the associated molecular mechanism(s). Our results showed that Nic potentially inhibited PC cell lines growth, colony formation, and arrest cell cycle progression. Further, mitochondrial stress coupled with mitochondria fragmentation is implicated in the activation of the apoptosis pathway, and inactivation of mTORC1 resulted in autophagy leads to Nic-mediated cell death. More importantly, at the molecular level, potential effects of Nic on PC cells are mediated through the stimulation of Gsk3β–phosphorylation that leads to inhibition of hedgehog and mTORC1 signaling pathways. Intriguingly, we observed that p-Gsk3β regulated cellular response was independent of β-catenin activation. Additionally, we found evidence that p-Gsk3β is an accountable factor for inhibition of non-canonical Hh/Gli cascade via upregulation of Sufu and Gli3 as well as stimulation of autophagy for Nic-induced PC cell death. 

## 2. Materials and Methods

### 2.1. Chemicals and Reagents 

Roswell Park Memorial Institute (RPMI)-1640 Medium (R8758-500 ML), Dulbecco’s Modified Eagle’s Medium (DMEM)—high glucose (D5796-500 ML), Iscove’s Modified Dulbecco’s Medium (IMDM) (SH30228.01), Dulbecco’s Phosphate Buffered Saline (SH30028.02), Keratinocyte-SFM (1×) (17005-042-500 ML) containing Keratinocyte-SFM Supplement: Human Recombinant Epidermal Growth Factor and Bovine Pituitary Extract (37000-015), HyClone Trypsin (SH30042.01), LY 294002 (440202), 2′,7′-Dichlorofluorescin diacetate (D5883) and propidium iodide (PI) (537059) were purchased from Sigma-Aldrich, St. Louis, MO, USA. Monodansylcadaverine, (MDC) (15571) was purchased from Cayman Chemical, Ann Arbor, MI, USA. *N*,*N*,*N*′,*N*′-tetramethylethylenediamine (TEMED) (AC138450500) and β-mercaptoethanol from Acros organics. Tween 20 (BP337) was purchased from Fisher Bioreagent. Pierce™ ECL Western Blotting Substrate (32106), MitoSOX Red (M36008), and MitoTracker Deep Red FM (M22426) were purchased from Thermo Fisher Scientific, Waltham, MA, USA. Primary and secondary antibodies were used in Western blotting and immunofluorescence listed in Appendix A. All other reagents were of the highest grade and commercially available. 

### 2.2. Cell Lines and Culture

Human pancreatic cancer cell lines (SW1990, COLO 357, T3M4, CFPAC1, CD18/HPAF, BxPC3, MIA PaCa, PANC-1, AsPC-1, and Capan-1), and normal pancreatic cell lines (human pancreatic nestin expressing (HPNE), and normal pancreatic ductal (HPDE) cells) were obtained from American Type Culture Collection. All the cell lines were tested for mycoplasma contamination before use and validated by short tandem repeat (STR) DNA profiling University of Nebraska Center (UNMC). Human pancreatic cancer cell lines and normal pancreatic cell lines were maintained in DMEM and RPMI, respectively, supplemented with 10% fetal bovine serum (FBS) and antibiotics (100 U/mL penicillin and 0.1 mg/mL streptomycin) at 37 °C with 5% CO_2_ in a humidified atmosphere. Further, we also used human pancreatic cancer-associated fibroblast cell lines like 09-11,09-17,10-32 [30], 10-15,10-03 CAFs) and normal fibroblast (09-26N) in this study.

The SW1990 cell line (moderately differentiated) was established from a spleen metastasis of a grade II pancreatic adenocarcinoma [31,32]. Whereas the COLO 357 cell line (well-differentiated) was developed from a lymph node metastasis of a human pancreatic adenocarcinoma and reported to express acinar digestive enzymes with unique karyology and allozyme profile [33,34,35]. Moreover, poorly differentiated cell lines such as PANC-1, AsPC-1, MIA PaCa-2, and well-differentiated cell line Capan-1 were also utilized to evaluate the therapeutic efficacy of Nic on PC.

### 2.3. Cell Viability Assay

MTT (3-(4,5-dimethylthiazole-2-yl)-2,5-dipenyltetrazolium bromide) assay: Pancreatic cancer and normal pancreatic cells were seeded (2.5 × 10^3^ cells/well) into a 96-well plate and then incubated with different dilutions of Nic or LY294002 (Gsk3β activator) for different time points. After incubation, MTT was added in a final concentration of 0.5 mg/mL for 3 h. After that, supernatants were discarded, and 100 μL of DMSO was added to each well, and the absorbance of the dissolved formazan crystals formed inside the viable cells was measured. The optical density (OD) was measured at a wavelength of 570 nm. As per the requirement of the experiment, PC cells were also treated with spautin-1(autophagy inhibitor) followed by treatment of Nic at different time points (24 h, 48 h, and 72 h) and a cell survival assay was performed as described above.

Calcein-AM assay: Cells were seeded into 96-well plates and then incubated with Nic alone or with/without autophagy inhibitors for different time points as described in an earlier section. Following incubation with a non-fluorescent, hydrophilic compound calcein-AM (structure A), the cells were washed with phosphate-buffered saline (PBS), decreasing the possibility of background fluorescence from the non-specific extracellular binding and the hydrolysis of calcein-AM in an aqueous solution. Finally, fluorescence was measured after 30 min incubation in constant intervals (5 min) at an excitation and emission wavelength of 485 and 528 nm, respectively, using the FLUOstar for up to 60 min. Maximum fluorescence values of each well were employed for the construction of concentration-response curves. 

### 2.4. Annexin-V/Propidium Iodide Labeling and Flow Cytometry Assay for Apoptosis

Apoptosis was measured by Annexin-V/Cy™5 (BD Biosciences) and propidium iodide (Roche Diagnostics) staining as per the manufacturer’s protocol. Briefly, cells (1 × 10^6^ cells/mL) were cultured in 6-well plates and treated with Nic alone for different periods (24 h, 48 h, and 72 h) or subjected to Nic in combination with or without Gsk3β activator (LY-294002) or autophagy inhibitor (spautin-1) for 24 h. Adherent cells were harvested, washed twice with PBS, and resuspended in 1× calcium-binding buffer, and double-stained with Annexin-V/Cy™5 and propidium iodide for 20 min at 37 °C. Cells were analyzed immediately by using a FACS Canto™ flow cytometer.

### 2.5. Cell Cycle Analysis

Briefly, cells were seeded (1 × 10^6^ cells) and synchronized by double thymidine block and treated with Nic (10 µM) for 24 h followed by fixing in 70% ethanol overnight at 4 °C. Later, the cells were washed with PBS and stained with Telford reagent (50 μg/mL propidium iodide, 90 mM EDTA, 0.1% Triton X-100, and 1μg/mL RNase A) for 1 h at 4 °C. The DNA content of stained cells was analyzed by using a FACS Canto™ flow cytometer.

### 2.6. Western Blotting

For Western blot analysis, control and treated cells from different groups and treatment time points were harvested and washed thoroughly with 1× cold PBS. Cell lysates were prepared by extracting the proteins by using radio-immuno precipitation assay (RIPA) buffer (50 mM Tris-HCl, pH 7.4; 1% NP-40; 1% sodium deoxycholate; 0.1% sodium dodecyl sulfate (SDS), 150 mM NaCl; 2 mM EDTA; 25 mM sodium fluoride, sodium orthovanadate, and sodium fluoride) supplemented with 1× protease inhibitor cocktail (Roche) and 1 mM phenylmethylsulfonyl fluoride (PMSF) and kept on ice for at least 30 min followed by syringe passage for complete lysis. The lysates were clarified by centrifugation at 12,000× *g* for 20 min at 4 °C, and collected in fresh Eppendorf tubes. The protein concentration of the extracted lysates were measured by bicinchoninic acid (BCA) method; an equal quantity of total protein (20–30 µg) per lane were loaded and separated by sodium dodecyl sulfate-polyacrylamide gel electrophoresis (SDS-PAGE) using laemmeli buffer (Tris 6.8, 10%SDS, 100% glycerol, Bromophenol blue, and β-mercaeptoethanol) and transferred to polyvinylidene fluoride (PVDF) membranes. The membranes were blocked with non-fat dry milk with phosphate-buffered saline with 0.1% Tween 20 (PBST) for 2 h at room temperature and then incubated with primary antibodies in 0.5% bovine serum albumin (BSA)/PBS overnight at 4 °C. After overnight incubation, the membranes were washed twice with PBST for 10 min each, and then incubated with respective secondary peroxidase-conjugated antibodies for 1 h at room temperature (RT) and developed using chemiluminescence reagent. Finally, the detection was performed with an enhanced chemiluminescence detection system (iBright, Thermofisher, or X-ray films). After developing, the membrane was stripped and re-probed using another primary antibody of interest or β-actin to confirm equal loading. Each experiment was repeated three times to assess for consistency of the results. The whole western blot can be found at Appendix A.

### 2.7. Colony Formation Assay

Cells were seeded in triplicates at a density of 1000 cells/well in 6-well plates. Cells were grown and treated with Nic ~IC_20_ (5 μM); ~IC_50_ (10 μM) for 48 h, then replaced with fresh medium, allowed to foster for 2 weeks in a humidified atmosphere (95% humidity) at 37 °C and 5% CO_2_, to form colonies, and finally stained by crystal violet and photographs were taken. Cell colony was then dissolved in 10% glacial acetic acid and the optical density was measured at 590 nm under a microplate reader. The cell colony was then dissolved in 10% glacial acetic acid, and the optical density was measured at 590 nm under a microplate reader. 

### 2.8. Cell Migration Assays

Cells (1 × 10^6^) were seeded on the upper chamber of an insert (8-mm pore size; BD Bioscience) containing serum-free medium for cell migration, and 20% FBS in DMEM was added in the lower chamber of the 6-well plate. After overnight incubation (16 h), migrated cells in the lower chamber were stained (Diff-Quik Stain Set, Siemens Healthcare Diagnostics, Inc., Newark, DE, USA). Images were captured using an EVOS FL Auto Imaging System at 10× magnification (EVOS FL Auto imaging system, Life Technologies, Thermo Fisher Scientific, Waltham, MA, USA).

### 2.9. Measurement of Intracellular and Mitochondrial Reactive Oxygen Species (ROS) Level

The intracellular reactive oxygen species (ROS) levels were measured using 2,7-dichlorodihydrofluorescein diacetate (DCFH-DA), which is oxidized into highly fluorescent 2,7-dichlorofluorescein in the presence of intracellular ROS. Mitochondria-specific ROS was determined using MitoSOX red reagent COLO 357 and SW1990 cells were seeded in 96-well plates and treated with Nic for 2 h. After that, cells were collected and incubated with 2′,7′-dichlorofluorescin diacetate (DCFDA) (10 µM for 30 min) or MitoSOX (5 µM for 30 min) in the dark and rinsed twice with PBS and transferred to the black well plate. Fluorescence was measured at absorbance 485/535 nm for DCFDA stained cells and 510/580 nm for MitoSOX stained cells using a fluorescent plate reader. The results were expressed as fluorescence intensity units relative to the corresponding control (untreated) cells.

### 2.10. Confocal Microscopy

To detect nuclear translocation of β-catenin, Gli1, and Sufu, COLO 357 and SW1990 cells were seeded on coverslips in a 12-well plate and treated with vehicle or Nic (10 µM) for 24 h and 48 h. Cells were then fixed in 4% paraformaldehyde and permeabilized with the 0.1% Triton X-100. Cells were washed with PBS and blocked with 2% BSA and incubated with the required antibody overnight, followed by 1 h incubation with fluorescence-tagged secondary antibodies (FITC, Cy-3), then mounted on slides with VECTASHIELD-containing 4′,6-diamidino-2-phenylindole (DAPI) (Vector Laboratories, Burlingame, CA, USA). Images were captured at 63× using a confocal laser scanning microscope, Carl Zeiss, LSM 800 META (CLSM). Next, to detect the co-expression of p-Gsk3β and Gli-1, cells were treated with Nic alone or subjected to Nic with or without Gsk3β activator (LY-294002) or autophagy inhibitor (spautin-1) for 24 h. Moreover, the formation of autophagosomes in treated cells was demonstrated by the localization of LCI/II. Cells were processed as described above with slight modification.

### 2.11. Mitochondria Imaging

Mitochondrial morphology was visualized using Mito Tracker Red as molecular probes [36], which passively diffuses across the plasma membrane and accumulates in active mitochondria in cells. Briefly, cells were grown on coverslips in a 12-well plate and treated with Nic for 24 h. Cells were then stained with 100 nM of MitoTracker Red for 30 min at 37 °C in the dark, washed with PBS three times. Further, cells were fixed with 4% paraformaldehyde and counterstained with VECTASHIELD-containing DAPI, and images were captured at 63× using Carl Zeiss microscope (LSM 800 META).

### 2.12. Monodansylcadaverine (MDC) Staining

A fluorescent compound, MDC, has been proposed as a tracer for autophagic vacuoles [37]. Monolayers of cells were cultured on coverslips in a 12-well plate and treated as indicated. Slides were washed with a culture medium without serum. Cytoplasmic vacuoles were stained with MDC according to the method described [37]. Briefly, cells were exposed to 50 mM of MDC for 20 min at 37 °C. After that, cells were washed with 4× PBS for removal of unbound MDC and immediately analyzed by fluorescence microscopy. Images were captured at 10×; 20× magnification. At least 5 areas per well were analyzed.

### 2.13. Fluorescence Imaging

COLO 357 and SW1990 cells were grown on coverslips in a 12-well plate and treated with Nic alone at a different time and concentration-dependent or incubated with vehicle, Nic, spautin-1, or in combination for 24 h as per experimental requirements. Following cells were incubated with calcein-AM (1 µM for 30 min) or DCFDA (10 µM for 30 min). Cells were processed as described earlier Section 2.11. Images were grasped at 10×, 20× using EVOS FL Auto Imaging System (EVOS FL Auto, Life Technologies, Carlsbad, CA, USA).

### 2.14. Nuclear and Cytoplasmic Extraction

In order to analyze the subcellular localization of various molecules upon Nic treatment, nuclear and cytoplasmic fractions were isolated by using suitable nuclear/cytoplasmic extraction buffers supplemented with a protease inhibitor cocktail. Briefly, cells were scrapped from the culture dishes, washed twice in 1× PBS and lysed for cytoplasmic fraction using harvest buffer containing 10 mM HEPES pH 7.9, 50 mM NaCl, 0.5 M sucrose, 0.1 M EDTA and 0.5% triton X 100. Supernatant containing cytoplasmic proteins were separated by centrifugation at 1000 rpm for 10 min at 4 °C. Nuclei pellets were further washed extensively with Buffer A (10 mM *N*-(2-Hydroxyethyl)piperazine-*N*′-(2-ethanesulfonic acid) (HEPES) pH 7.9, 10 mM potassium chloride (KCl), 0.1 mM ethylenediaminetetraacetic acid (EDTA), and 0.1 mM ethylene glycol-bis(β-aminoethyl ether)-*N*,*N*,*N*′,*N*′-tetraacetic acid (EGTA)) and lysed by using Buffer C (10 mM HEPES pH 7.9, 50 mM sodiun chloride (NaCl), 0.1 mM EDTA, 0.1 mM EGTA and 0.1% nonidetP-40) following vigorous shaking for 15 min at 4 °C. Finally, both cytoplasmic and nuclear fractions were centrifuged at top speed, i.e., 14,000 rpm for 15 min at 4 °C and subjected to Western blot analysis as mentioned in Section 2.6. 

### 2.15. Immunoprecipitation

Beclin-1 and BCL2 interaction was assessed by performing the co-immunoprecipitation assay. Briefly, samples (1 mg/mL each) were pre-clarified by incubating with protein A/G plus agarose beads (Santa Cruz Biotechnology, Dallas, TX, USA) for 1–2 h followed by washing and centrifugation. Pre-cleared lysates were incubated with desired antibodies (beclin-1 and isotype control antibodies) along with 30 μL of protein A/G plus agarose beads overnight at 4 °C with continuous agitation. After that, immune complexes were centrifuged at 3000 rpm for 3 min to remove supernatant, the pellet was washed five times with RIPA lysis buffer (four times for 3 min), and co-immunoprecipitated proteins were assessed using 12% SDS-agarose gel electrophoresis along with input (3–5% of the total protein) and IgG control samples in Laemmli buffer. For co-immunoprecipitation, protein samples were resolved in 12% SDS-PAGE, transferred onto a PVDF membrane, probed with primary anti-BCL2 or anti-Beclin1 antibody, and incubated at 4 °C overnight after blocking with 5% skimmed milk in PBST. Next, the membrane was washed with PBST (three times for 10 min), probed with the secondary antibody for 1 h at RT, and visualized with chemiluminescence ECL reagent. 

### 2.16. Statistical Analysis

We carried out statistical analysis using Prism 7.0b (GraphPad Software Inc., San Diego, CA, USA) statistical analysis software. Data are expressed as mean ± standard error of the mean (SEM) for at least three separate determinations for each experiment unless otherwise specified. Statistical significance was determined by one-way analysis of variance (ANOVA) and Newman–Keul’s test for multiple groups comparison. For two-group comparisons, depending on data distribution, a two-tailed, unpaired Student’s *t*-test was used. *p* Values less than 0.05 were considered statistically significant.

## 3. Results

### 3.1. Nic Specifically Suppresses Pancreatic Cancer Cell Growth and Migration

In order to analyze the efficacy of Nic for PC therapy, we performed cell viability assay on various PC cell lines (SW1990, COLO 357, and T3M4) by treating them with different concentrations of Nic for various time points, and the cell viability was observed by examining mitochondrial activity using the MTT uptake assay. We observed that Nic significantly reduced the viability of PC cell lines in a dose-dependent manner with the half maximal inhibitory concentration (IC_50_) of ~10 μM (*p* < 0.001). Interestingly, similar concentrations of Nic was ineffective in inhibiting the growth of normal HPNE cells and normal HPDE cells (Figure 1A,B). Therefore, our results demonstrated that Nic has an anti-proliferative/anti-cancerous effect on PC cells without affecting the normal pancreatic cells. However, aggressive PDAC usually consists of cells with different differentiated status. To mimic this, we also analyzed the inhibitory effect of Nic on PANC-1, AsPC-1, and MIA PaCa-2 cell lines (Appendix A). We observed that Nic has the ability to inhibit the cell growth of PC cells with different differentiation statuses ranging from moderate, well, and poorly differentiated PC cells. Hence, to delineate the underlying mechanism, the effective concentration of Nic, i.e., ~IC_50_ (10 μM) 50% mortality; ~IC_20_ (5 μM) 20% mortality, was used in subsequent experiments. Importantly, *KRAS* drives 32% of lung cancers, 40% of colorectal cancers, and 85% to 90% of pancreatic cancer [38]. Furthermore, it has also been reported that niclosamide inhibits Ras-driven oncogenic transformation via activation of Gsk-3 in colorectal cell lines [39]. In this study, we have selected SW1990 and COLO 357 cell lines to decipher the underlying mechanism of Nic. Both cell lines are well-defined cell lines (SW1990 contain mutant *CDKN2A*, *KRAS*, heterozygous for TP53 p.Pro191del; and COLO 357 contain mutant *KRAS* and SMAD4) derived from pancreatic adenocarcinoma (PDAC), showed potential tumorigenic ability in athymic nude mice and are widely used in pancreatic cancer research [40,41,42].

This anti-cancerous effect of Nic was further validated in colony formation and trans-well migration assays and observed that Nic significantly reduced in-vitro colony formation ability (Figure 1C) and migratory potential (Appendix A) of PC cells compared to untreated control cells. The dose-dependent inhibition of PC cell migration and colony formation ability was observed. In contrast, the treatment of PC cells with a 5 µM concentration of Nic led to the suppression of clonogenic and migratory potential. However, its effect was prominent, i.e., complete inhibition of colony formation and migration at 10 µM concentration of Nic. Additionally, we also observed the downregulation of mesenchymal markers, i.e., *N*-cadherin, no effect on the expression of epithelial marker E-cadherin, and claudin-1 in COLO 357 and SW1990 cells treated with Nic (5 µM and 10 µM) (Appendix A) leading to inhibition of migration potential of PC cells. Next, we demonstrated the inhibitory efficacy of Nic on different pancreatic cancer-associated and normal fibroblasts. Compared to the normal pancreatic fibroblasts, a significant reduction in the viability of pancreatic cancer associated-fibroblast cells (CAFs) was observed after 24 h incubation with Nic (5 µM, 10 µM), and this effect was more prominent after 48 h treatment. (Appendix A). We then investigated the effect of Nic on fibroblast activation. The reduction in Alpha smooth muscle actin (α-SMA), a fibroblast activation marker, upon Nic treatment was found. However, no change in the expression of fibroblast activation marker (FAPα) suggested that Nic treatment reduces the viability of CAFs without altering their activation status (Appendix A).

Considering the effect of Nic in reducing cell viability, we performed a cell cycle analysis. After treatment with Nic (IC_50_), the cell distribution was significantly altered in all cell cycle phases compared to the control cells. Further, an increased accumulation of PC cells in the G1-phase of the cell cycle (Figure 1D) in Nic treated group compared with untreated control group. Together, these results show the anti-proliferative effect of Nic, specifically on PC cell growth, and significantly impacts their migratory potential in vitro. 

### 3.2. Nic Increases Intracellular and Mitochondrial Reactive Oxygen Species and Promotes Mitochondrial Fragmentation Resulting in PC Cell Apoptosis

In order to delineate the cause of Nic-mediated PC cell death (COLO 357, SW1990), we first analyzed the apoptotic potential of Nic by assessing phosphatidylserine translocation using Annexin V-cy5/PI double staining followed by flow cytometry. We noticed that the proportion of the cells in the lower and upper right quadrant, which corresponds to early and late apoptotic cells, were significantly increased in a time-dependent manner, i.e., 24 h (*p* < 0.01); 48 h (*p* < 0.001); 72 h (*p* < 0.001) after Nic treatment (Figure 2A). This data was further confirmed by Western blot analysis, where we determined the expression levels of the apoptotic markers upon Nic treatment. The Western blotting results also showed that treatment of PC cells with Nic led to an increased expression of apoptotic proteins, including cleaved Poly (ADP-ribose) polymerase (PARP), cleaved caspase-9, and downregulation of BCL2 in a time-dependent manner (1 h, 3 h, 12 h, 24 h, and 48 h) (Figure 2B). Quantitative analysis of cleaved PARP, full-length PARP, and the ratio of cleaved PARP/full-length PARP showed a significant upregulation of cleaved PARP in Nic treated samples (Appendix A). We also found the significant upregulation of cleaved caspase-3 in COLO 357 and SW1990 PC cells treated with Nic (10 µM) for 24 h and 48 h. (Appendix A). Furthermore, we demonstrated the Nic-mediated apoptosis on poorly differentiated cell lines i.e., PANC-1 and AsPC-1(Appendix A) and 10–32 CAFs (Appendix A). A significant apoptosis was observed after Nic treatment (5 µM,10 µM). Altogether, these data suggested that Nic could suppress pancreatic cell growth via inducing apoptosis. 

Reactive oxygen species (ROS) are considered major drivers of apoptosis [38]. To analyze the effect of Nic in altering ROS, we evaluated the generation of intracellular and mitochondrial ROS upon Nic treatment via using DCFDA and MitoSox Red molecular probe, respectively. A dose-dependent increase in intracellular (Figure 2C) and mitochondrial ROS (Figure 2D) upon Nic treatment suggested the role of Nic–induced ROS generation in PC cells leading to apoptosis. As we observed a significant increase in mitochondrial ROS, which is well documented as a mitochondria dysfunction, i.e., mitochondria undergoing excessive fragmentation via mitochondria fission, is a crucial aspect for mitochondria-mediated cell death [43,44]. Our results demonstrated that Nic predominantly induces mitochondrial ROS, which leads to mitochondria fragmentation in a dose-dependent manner and triggers mitochondria-mediated apoptosis (Figure 2E). Mitochondria fission is regulated by dynamin-related protein1 (Drp1) [36,44]; therefore, mitochondrial fragmentation was further confirmed by increased DRP1 levels upon Nic treatment (Figure 2F) in a time-dependent manner. Taken together, our data suggested that Nic potentially induced mitochondrial stress via intracellular and mitochondrial ROS production resulting in mitochondrial fragmentation and hence promoting activation of the apoptotic pathway in PC cells.

### 3.3. Nic Stimulates Autophagy through Inhibition of mTORC1 Signaling in Pancreatic Cancer (PC) Cells

Altered mitochondrial dynamics are also connected with increased autophagy [45]. Increased mitochondrial ROS may directly lead to apoptosis, or it might induce autophagy, which further results in apoptosis [46]. Previous studies have also shown that Nic is a potent inducer of autophagy [47,48]. Therefore, to delineate the effect of Nic in inducing autophagy, first, we examined the autophagosome formation after exposure of PC cells with Nic by using auto-fluorescent probe MDC, a specific autophagolysosome marker [37]. The accumulation of MDC-labeled autophagic vacuoles with increased number and size were observed with varying concentrations of Nic treatment in PC cell lines (Figure 3A). We also noticed the time-dependent upregulation of LC3B-II/LC3B-I (Figure 3B) and an increased ratio of LC3B-II/LC3B-I (Appendix A). Further, we observed diffused cytoplasmic staining of LC3-II in control cells, whereas LC3-II puncta formation was increased in Nic treated PC cells (Figure 3C), validating autophagy induction upon Nic treatment. Beclin-1 is a crucial mediator of autophagy and forms a multimeric complex necessary for autophagosome formation [49]. Hence, we next evaluated Beclin-1 and its downstream effector molecules upon Nic treatment (Figure 3D). Like the LC3-II/LC3-I ratio, an increased expression of Beclin-1 was also observed after Nic treatment in a time-dependent manner. Together, these results suggested the formation of autophagosomes upon Nic treatment. Autophagosome processing is dependent on autophagosome-lysosome fusion or lysosome function [50] and is inversely related to SQSTM1/p62 levels [51]. Increased LAMP2 (a lysosome membrane protein) expression with a concomitant decrease in SQSTM1/p62 (a protein that recruits ubiquitinated proteins to autophagosomes) levels further cemented Nic as an inducer of autophagy in the PC cells (Figure 3D). The mTOR is one of the major signaling pathways regulating autophagy in eukaryotic cells. It is a negative regulator of autophagy [52], and previous studies have demonstrated Nic-mediated mTORC1 inhibition via modulation of cytoplasmic pH [53]. Activation of mTORC1 induces the phosphorylation of p70S6 kinase (p70S6K) and eukaryotic initiation factor 4E (eIF4E) binding protein 1 (4E-BP1), leading to the enhanced translation of a subset of mRNAs that are critical for cell growth [54]. Based on this information, we further investigated the effect of Nic on mTORC1 signaling in PC. Interestingly, Nic treatment abolished the phosphorylation of p70S6K1 at Thr421/Ser 424 and phosphorylation of 4E-BP1 at Ser65 in a time-dependent manner (Figure 3E), suggesting that Nic stimulates autophagy in PC cells by inhibiting mTORC1 signaling.

### 3.4. Nic-Induced Autophagy Leads to PC Cell Death by Disrupting the Beclin1–BCl2 Interaction

To test whether Nic-induced autophagy is a cause of PC cell apoptosis, we used spautin-1, a selective autophagy inhibitor [55]. We determined the effect on cell survival in different treatment groups, i.e., vehicle control, Nic, spautin-1, and a combination of Nic and spautin-1. We observed increased cell viability in Nic and spautin-1 combination group with respect to the Nic alone treatment (Figure 4A). Interestingly, this protective effect was significantly displayed after 48 h incubation with Nic, and a more prominent effect was observed at 72 h. Concomitantly, we validated these findings using calcein-AM assay, a non-fluorescent, hydrophilic compound that quickly permeates intact live cells [56].

Interestingly, an intense calcein-AM fluorescence due to more viable cells present in Nic and spautin-1 treated combination group was observed compared to the Nic-alone treated group. This observation provided evidence that attenuation of autophagy by using spautin-1 almost rescued the growth inhibitory effect of Nic and displayed intense fluorescence like control group (Figure 4B). Moreover, these results were confirmed by measuring the quantitative fluorescence of calcein-AM as determined by fluorescence spectrophotometry, and a similar observation was apparent (Appendix A). These results revealed that Nic–mediated autophagy resulting in PC cell death. These observations were further corroborated by Annexin V/PI staining of PC cells under similar conditions showing pretreatment of PC cells with spautin-1 rescued Nic induced apoptotic effect resulting in higher live cell population as compared to Nic alone-treated group (Figure 4C). Similarly, treatment of PC cells with Nic and spautin-1 combination therapy displayed lower intracellular ROS levels than the Nic alone treated group, ensuring that Nic induced autophagy is a major driver of PC cell apoptosis. To further comprehend the relation between apoptosis and autophagy at the molecular level upon Nic treatment, we again performed functional blockage of autophagy using spautin-1 and determined its effect on autophagy as well as apoptosis markers. The decreased expression of LAMP2, Beclin-1, and increased expression of BCL2 was found in spautin-1 pretreated PC cells incubated with Nic compared to Nic alone treated cells (Figure 4E,F). These results suggested that autophagy inhibition suppressed the Nic-mediated apoptosis that apparent by upregulation of BCL2 expression and downregulation of Beclin-1 expression in Spautin-1+ Nic treated PC cells.

Next, to decipher the underlying mechanism of autophagy and apoptosis crosstalk, we performed a Co-Immunoprecipitation (Co-IP) assay and examined the effect of Nic on the physical binding of Beclin-1 with BCL2 in PC cell lines treated with Nic for various time point (24 h & 48 h). Since Nic treatment decrease the expression of Bcl2, there is very low or no availability of BCL2 to form a complex with Beclin-1; hence the absence of Beclin-1-Bcl2 complex results in the induction of autophagy-mediated apoptosis in PC cell lines (Figure 4G).

### 3.5. Nic Promotes Gsk3β Inactivation and Inhibits Hh/Gli Cascade via Upregulation of Sufu and Gli3, a Negative Regulator of Hh Signaling in PC Cells

Recent studies have shown the potential role of Gsk3β in promoting autophagy in PC [57]. Several studies have shown that Nic potentially inhibits Wnt/β-catenin signaling in multiple cancers, including ovarian, colorectal, breast, glioma, renal, head, and neck carcinoma cells [16,17,18,19,20,21,22]. Along similar lines, to decipher the mechanistic effect of Nic treatment on Gsk3β status and autophagy, first, we analyzed the expression pattern of members of Wnt/β-catenin signaling pathway, i.e., Gsk3β, p-Gsk3β, p-β-catenin (active) in normal (HPNE) vs. PC cell lines (T3M4, CFPAC1, SW1990, COLO 357, CD18/HPAF, BxPC3, MIA PaCa 2, and Capan-1). Almost all the PC cell lines showed substantial Gsk3β expression (Figure 5A). However, SW1990 and COLO 357 were selected for further analysis based on significant expression of Gsk3β in the subsequent experiment. To analyze the effect of Nic treatment on the Wnt/β-catenin pathway, first, we analyzed the expression of Gsk3β, p-Gsk3β (Ser-9), p-β-catenin (Ser-33/37Thr 41) (active), and total-β-catenin upon Nic treatment in a time-dependent manner. Unlike p-Gsk3β (Ser-9) expression, which showed a gradual increase upon Nic treatment with time, we did not observe any change in p-β-catenin expression (Figure 5B) as well as its nuclear localization upon 48 h of Nic treatment (Figure 5C). Similarly, there is no change in p-β-catenin expression in nuclear and cytoplasmic fractions upon different concentrations of Nic treatments (Figure 5D). However, in line with the literature [19], we observed dose-response change in p-β-catenin expression upon Nic treatment in the colorectal cell line, LS-180 (Appendix A). These results revealed that the regulatory response of p-Gsk3β is independent of β-catenin and suggested that the different downstream mediator of Gsk3β inactivation is implicated in Nic treatment, particularly in PC. Gsk3β has been shown to regulate Hedgehog (Hh) signaling [58,59,60], and aberrant/constitutive activation of the canonical as well as non-canonical Hh signaling co-exists in PC and other cancers [61,62]. Therefore, we analyzed the effect of Nic treatment on Hh signaling effectors expression levels such as Sufu, Gli1, and Gli3 (Figure 6A). Interestingly, unlike untreated controls, substantial upregulation of Sufu and Gli3 with concomitant downregulation of Gli1 expression (Figure 6A) suggested the involvement of a non-canonical regulatory mechanism where components outside the Hh-Ptch-Smo-Gli paradigm plays a crucial role in the activation of Gli transcription [62]. To further confirm this non-canonical paradigm, we demonstrated the effect of Gsk3β activation on the nuclear and cytoplasmic expression of Hh signaling effectors upon Nic treatment. We observed increased accumulation of Gli3 in nuclear fraction with increased expression of p-Gsk3β, Sufu in both cytoplasmic and nuclear fractions, with concomitant downregulation of Gli1 expression (Figure 6B). Similarly, we observed increased nuclear accumulation of p-Gsk3β with reduced expression of Gli1 in confocal microscopy (Figure 6C). Simultaneously, decreased expression of Shh, i.e., ligand/morphogen of Hh signaling [63], indicated the inactivation of Hh signaling after Nic exposure in PC cells (Appendix A), which was further evident by nuclear translocation of Sufu upon Nic treatment (Figure 6D). For final validation, we demonstrated the effect of Nic on well-known target molecules of Gsk3β such as p21^waf1/cip1^, p27^kip1,^ and cyclin D1 [64,65,66] as well as Hh pathway downstream molecule, i.e., p-cMyc (s-62) [67]. Our Western blot analysis revealed that increased expression of p21^waf1/cip1^, p27^kip1^, and reduced expression of cyclin D1 and p-cMyc were observed after Nic treatment in a time-dependent manner (Figure 6E). These results altogether indicated that Nic-mediated phosphorylation of Gsk3β supports the inhibition of the Hh pathway by attenuation of Gli1 (an activator transcription factor) through stimulation of the Gli3 (an inhibitory transcription factor) with upregulation of Sufu (a negative regulator) in PC cells.

### 3.6. Nic Induced Gsk3β Inactivation Inhibits Gli1 Activation, Negatively Impacting Hh Signaling and Promoting Autophagy-Mediated Apoptosis

It is important to note that the oncogenic role of Gli1, a key transcription factor of Hh signaling, is primarily modulated via canonical and non-canonical signaling pathways [61,62]. Gsk3β acts as both a positive and a negative regulator of the Gli1 activation, which further decides the fate of cells either in survival or apoptosis [58,59,60]. Our results indicated that the involvement of non-canonical mechanism of Nic mediated cell death, we used a selective activator of Gsk3β, i.e., LY-294002, that induces reactivation of Gsk3β by promoting its dephosphorylation on the Ser-9 [68]. The effect of Gsk3β activation on PC cell survival was analyzed by subjecting the cells to the LY-294002 treatment [57]. In SW1990 cells, LY-294002-induced significant growth inhibition at 30 μM concentration (*p* < 0.001) (Figure 7A). Next, the outcome of activation of Gsk3β on the Hh pathway was determined by treating PC cells with LY-294002 and increased expression of Gsk3β, Sufu with decreased expression of p-Gsk3β, Gli3 without any effect on Gli1 expression were observed. In contrast, Nic -treated PC cells showed reduced expression of Gsk3β and increased expression of p-Gsk3β, Gli3, and Sufu with reduced Gli1 expression. Remarkably, in comparison with control (vehicle), LY-294002 and Nic treated PC cells showed a complete abrogation of Gli1 expression with decreased expression of p-Gsk3β and Gsk3β in PC cells treated with LY-294002 in combination with Nic treatment (Figure 7B). Additionally, these results were further confirmed by confocal microscopy by evaluating the nuclear and cytosolic expression of Gli1 protein and we observed decreased expression of Gli1 and p-Gsk3β in PC cells treated with Nic alone or in the combination with LY-294002 (Figure 7C). These findings suggested that Nic–mediated phosphorylation of Gsk3β is vital for attenuation of Gli1 activation; consequently, our results revealed that p-Gsk3β acts as an inhibitory regulator of Hh/Gli1 cascade in PC cells. Furthermore, we analyze the involvement of this pathway in autophagy-mediated cell death by treating, PC cells with Gsk3β activator (LY-294002), Nic, or a combination of Nic and LY-294002, and examined for expression of autophagy mediators or markers. A fluorescence study via MDC dye revealed that activation of Gsk3β via LY-294002 leading to the absence of MDC-labeled vacuoles (Figure 7D). In contrast, a significant number and sizes of MDC-labeled vacuoles were observed in Nic and combination with LY-294002. Furthermore, these results were also confirmed by detecting expression of autophagy markers (LCI/II and Beclin-1) using Western blot analysis. We observed similar results that Nic-mediated inactivation of Gsk3β promotes autophagy in PC cells (Figure 7E). Mechanistically, these findings suggested that Nic–mediated phosphorylation of Gsk3β is vital for attenuation of Gli1 activation resulting in inhibition of Hh signaling, consequently leading to growth suppression and promoting autophagy-mediated PC cell deaths.

## 4. Discussion 

Recurrence and mortality are a huge challenge for PC treatment [6]. Although new drugs, combination therapies, and/or immunotherapies are undergoing clinical trials, the impact of above therapies on PC patient survival dismal. Therefore, there is an unmet clinical need for alternative treatments to improve the outcome of patients with PC [7,8,69]. Drug repurposing is a common practice to develop the most promising new therapies for numerous diseases, including cancer [9]. Recent studies have reported the anti-cancerous effect of Nic, and its efficacy was being assessed on prostate cancer and colorectal cancer in different clinical trials [38,70]. Based on its gastrointestinal targeting efficacy, in the present study, we dissected its therapeutic utility in PC in vitro, focusing primarily on deciphering the impacts of Nic on various molecular regulatory mechanisms.

Our findings suggested that Nic exhibited an anti-proliferative effect on PC cells without affecting the normal pancreatic cells HPNE and HPDE by exerting G1 phase cell-cycle arrest, inhibition of colony formation, and migration potential PC cells. In addition to the above, we investigated both apoptotic and autophagy cell death mechanisms involved in Nic-induced cell death. It is well documented that apoptosis is also governed by ROS imbalance, and mitochondria serve as a major source of ROS [43,44]. Previous studies also reported that change in ROS level was associated with the intrinsic apoptosis [43] controlled by BCL2 family proteins. Our results demonstrated downregulation of anti-apoptotic marker BCL2 and upregulation of apoptotic markers, including cleaved PARP and cleaved caspase 9 upon treatment of PC cells with Nic in a time-dependent manner. Furthermore, the significant cause of Nic-mediated apoptosis in PC cells was perceived due to elevated intracellular and mitochondrial reactive oxygen species (ROS) coupled with mitochondria fragmentation. Accumulating evidence has indicated that mitochondria fragmentation via mitochondria fission was regulated by mitochondria fission protein dynamin-like GTPase (Drp1), crucial for mitochondria-mediated cell death.

Moreover, Nic has been identified as a potent inducer of mitochondria fission in HeLa cells [71]. Here, our results provided further evidence that Nic-induced apoptosis through disruption of mitochondria morphology via an increased level of ROS and induced expression of Drp1 by activating the mitochondrial intrinsic pathway. Simultaneously, autophagy induction was evidenced by the accumulation of MDC-labeled autophagic vacuoles with increased number and size in Nic-treated PC cells compared to untreated controls. Our study confirmed at a molecular level, that Nic mediated increased expression of autophagosome markers such as the lapidated form of LC3B, Beclin-1, and auto-phagolysosome fusion marker, i.e., LAMP2 in PC cells suggested its effect on autophagy. Autophagy is a dynamic process, and autophagosome turnover is an important aspect; therefore, we determined autophagosome turnover by measuring the adaptor protein sequestosome1 (SQSTM1/p62) [72]. SQSTM1/p62 is a protein that recruits ubiquitinated proteins to autophagosomes, and its expression levels inversely correlate with activation of autophagy [51]. The declined expression of p62 was apparent in Nic-treated PC cells in a time-dependent manner that anticipated the occurrence of the successful autophagy process.

Additionally, mTORC1 was a negative regulator of autophagy and involved during autophagy process, including the nucleation, elongation, maturation, and termination of the autophagosome. Moreover, previous studies have demonstrated that Nic-mediated mTORC1 inhibition via modulation of cytoplasmic pH [53]. In concurrence with the previous finding, we also found Nic-mediated autophagy was due to the inhibition of the mTORC1-dependent pathway that triggers autophagic flux in PC cells. These observations reinforced that mitochondrial stress and mTORC1-dependent autophagy are the predominant players of Nic-induced PC cell death.

Autophagy is not only distinguished as a type-II programmed cell death but also associated with cell survival. Previous studies have shown that autophagy is constitutively active and required for PDA growth [73]. Another point that needs to be considered is the role of Nic-mediated autophagy in PC cells resulting in survival or death. Thus, we treated PC cells either with Nic alone or together with a selective autophagy inhibitor spautin-1. Remarkably, we noticed that PC cells incubated with autophagy inhibitor in combination with Nic displayed a protective effect that resulted in PC cell survival. Further, co-treatment with spautin-1 attenuated the Nic-mediated apoptotic effect that was apparent by increased expression of anti-apoptotic protein BCL-2 as well as reduced intracellular ROS levels compared to Nic alone-treated PC cells. These observations indicated that Nic-mediated autophagy was leading to PC cell death. An emerging body of literature revealed that the autophagy-mediated cell death mechanism can occur either in the absence of detectable apoptosis signals or concomitantly with apoptosis [74]. Along these lines, we deciphered the underlying mechanism between autophagy and apoptosis and noticed that Nic involves in disrupting of interaction between autophagy regulating protein Beclin1 and anti-apoptotic protein BCl2 that causes apoptosis, which resulted in PC cell death. These results further corroborated that Nic–induced autophagy is significantly involved in the apoptosis of PC cells.

We observed the increased expression of p-Gsk3β (Ser9) in PC cells after Nic treatment. Surprisingly, the p-Gsk3 β-mediated regulatory response was independent of β-catenin modulation in PC, whereas its effect was β-catenin, dependent on colorectal cancer [19]. It has been reported that the anti-cancerous effect of Nic is mediated through altering different molecular signaling pathways in different cellular contexts. More precisely, Nic potentially inhibits the Wnt/β-catenin signaling pathway in adrenocortical carcinoma, ovarian cancer, glioma, colorectal cancer, breast, and renal cell carcinoma [16,17,18,19,20,21,22]. However, we observed β-catenin independent response via p-Gsk3β in PC; this unanticipated data may be due to different cancers and cell origins. Further, we also investigate the functional involvement of p-Gsk3β in the context of Hh signaling. Intriguingly, the non-canonical Hh/Gli cascade is implicated in Nic mediated inhibitory response in PC cells. Mechanistically, we demonstrated an increased expression of Gli3 and nuclear accumulation of Sufu, accompanied by decreased expression of Gli1 and Shh in Nic-treated PC cells. Therefore, upregulation of Sufu and Gli3 expression suggested the involvement of a non-canonical regulatory mechanism of Hh signaling that was further supported by decreased expression of Shh and Gli1, leading to Hh signaling inhibition. In non-canonical Hh signaling, the components signal was outside the Hh-Ptch-Smo-Gli paradigm and plays a crucial role in activating Gli transcription [62]. In general, Sufu, a negative regulator of Hh signaling [60], interacts with a conserved motif of the Gli protein and facilitates Gli phosphorylation by protein kinases such as PKA, Gsk3β, and CK1. Gli family members, including Gli1, Gli2, and Gli3, play a pivotal role in the activation and repression of Hh signaling [75]. Gli1 is a transcription activator of the pathway, while Gli2 and Gli3 act as activators or repressors depending on the signal and cellular context [76,77].

Cell cycle arrest at the G1 phase occurs via inhibition of Gsk3β with subsequent activation of p21^waf1/cip1^, p27^kip1^ [64,65], and our results showed the increased expression of p21, p27 in PC cells upon treated with Nic. These results indicated that Nic mediated inactivation of Gsk3β is also a plausible cause of G1-phase cell cycle accumulation. We also observed the decreased expression of Cyclin D1, cell cycle transition marker; generally, lack of Gsk3β-mediated phosphorylation of cyclin D1 suppresses their nuclear accumulation and degradation. Thus, we checked another Gsk3β-regulated protein, p-cMyc, a downstream Hh signaling target [67]. Our study showed decreased expression of p-cMyc, demonstrating the involvement of pGsk3β-mediated Hh signaling inhibition in Nic-treated PC cells.

Nevertheless, the crosstalk between Gsk3β and Gli-family members is well documented [58,59,62]. Either it can phosphorylate Gli proteins for degradation or truncated repressor forms in the inactivated state, with the help of Sufu and the release of Gli from the complex [59,60,77]. Notably, this study, for the first time, shows that the inactivation of Gsk3β supported the inhibition of Hh signaling via nuclear accumulation of Gli3 and Sufu. Hence, to prove an underlying non-canonical mechanism, we used a selective activator of Gsk3β, i.e., LY-294002, which induces reactivation of Gsk3β by promoting its dephosphorylation at Ser-9 [68]. Activation of Gsk3β by using LY-294002 showed increased expression of Gli1 and decreased expression of Gli3 while Nic-treated PC cells exhibited the increased expression of p-Gsk3β and reduced expression of Gli1 along with an increased expression of Gli3 in PC cells. Based on these observations, we speculated that Nic treatments in PC cells led to phosphorylation of Gsk3β predominantly at Ser 9 that enables the stabilization of Sufu and Gli3, thereby supporting Gli3 processing and eventually led to inactivation of Hh signaling.

## 5. Conclusions

In conclusion, our findings establish niclosamide (Nic) as a potential anti-cancerous drug for PC therapy. Nic-mediated inactivation of Gsk3β supports the modulation of the non-canonical Hh pathway and autophagy-mediated PC cell death (Figure 8). Nic-mediated cell death was associated with mitochondria-dependent apoptosis and mTORC1 regulated autophagy. Aberrant activation of Gsk3β is implicated in PC pathogenesis by altering the multiple oncogenic molecular signaling pathways [78]. A recent study showed that inhibition of Gsk3β sensitizes the PC cells to chemotherapy by DNA-damage response [79]. Therefore, targeting the inactivation of Gsk3β by using Nic represents an exciting approach for PC treatment. These findings provide the basis for an in-depth study of Nic in various PC animal models alone and in combination with other chemotherapeutic drugs, will provide valuable preclinical data for the inclusion of Nic for PC patient therapy in the future. Nevertheless, further efforts will be required to deliver Nic (oral bioavailability is very poor) and careful interpretation to determine the therapeutic potential of Nic in combination with chemotherapeutic drugs, the established standard of care therapies.

## Figures and Tables

**Figure 1 cancers-13-03105-f001:**
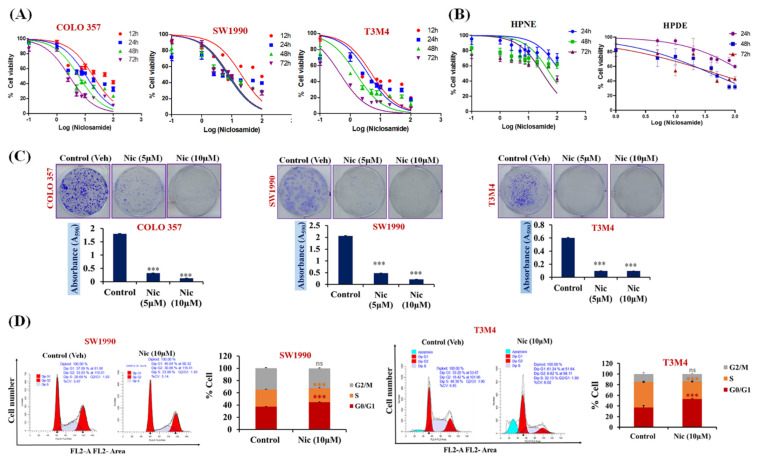
Nic-mediated inhibition of cell growth, colony formation, and G1-phase cell accumulation in pancreatic cancer (PC) cells. (**A**,**B**) Concentration (100 nM–100 µM) and time-dependent (12, 24, 48, and 72 h) effect of Nic treatment on the viability of PC cells (**A**) and normal (human pancreatic nestin expressing (HPNE), pancreatic ductal (HPDE)) pancreatic cell lines (**B**) were determined by MTT assay. Values are expressed as mean ± SEM (*n* = 5). (**C**) Evaluation of in vitro colony formation capabilities of PC cell lines (COLO 357, SW1990, and T3M4) treated with Nic ~IC_20_ (5 μM); ~IC_50_ (10 μM) (upper panel). Cell colony was then dissolved in 10% glacial acetic acid, and the optical density was measured at 590 nm under a microplate reader. Quantitative analysis of these colonies represented (lower panel) and values are expressed as mean ± standard error of the mean (SEM, *n* = 5), *p* values: *** *p* < 0.001 vs. control, Student’s *t*-test. (**D**) SW1990 and T3M4 cells-treated with Nic (10 μM) and stained with propidium iodide (PI) followed by flow cytometry to determine the cell cycle distribution based on DNA content (**left** panel). Quantitative analysis of these micrographs was shown as mean ± SEM, *p* values: *** *p* < 0.001 vs. control (**right** panel).

**Figure 2 cancers-13-03105-f002:**
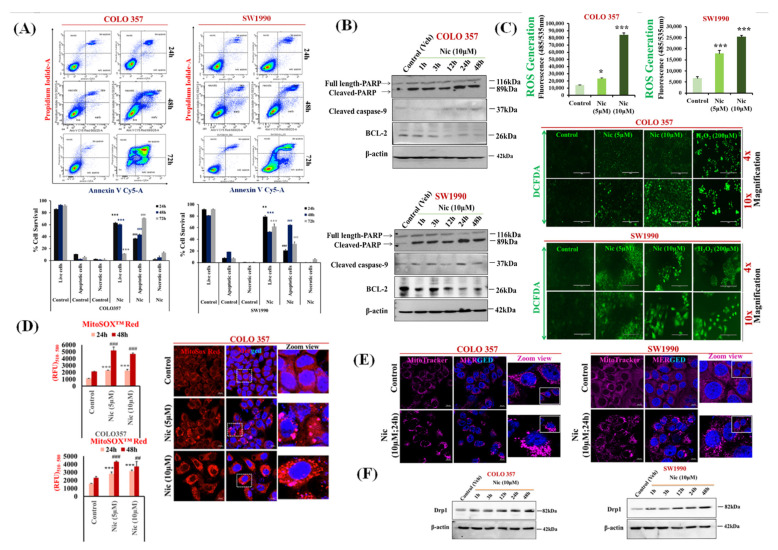
Nic increases intracellular and mitochondrial reactive oxygen species and promotes mitochondrial fragmentation resulting in PC cell apoptosis. (**A**) COLO 357 and SW1990 PC cells were treated with Nic (10 μM) for different time points (24, 48, and 72 h) and apoptosis was determined by flow cytometric analysis of annexin-V Cy-5/PI- dual stained cells (AV^+^/PI–intact cells; AV/PI^+^–nonviable/necrotic cells; AV^+^/PI and AV^+^/PI^+^—apoptotic cells) in PC cell lines, i.e., COLO 357 and SW1990 (upper panel). Quantitative analysis of these micrographs were shown as mean ± SEM (*n* = 3), *p* values: *** *p* < 0.001, ** *p* < 0.01 vs. live control cells; ^###^
*p* < 0.001, ^##^
*p* < 0.01 vs. apoptotic control cells (**lower** panel). (**B**) Representative Western blot images showing the expression of apoptotic markers such as cleaved-PARP, total PARP, cleaved caspase 9, and BCL-2 at different time points (1, 3, 12, 24, and 48 h) in COLO 357 and SW1990 cells treated with Nic (10 µM). β-actin is used as an internal loading control. (**C**) Intracellular ROS levels in PC cells treated with Nic (5 μM and 10 μM for 24 h) was evaluated using DCFDA fluorogenic dye (10 µM for 30 min) by measuring fluorescence at 485/535 nm (excitation/emission) using a fluorescent plate reader. The results were expressed as relative fluorescence intensity units. Values are expressed as mean ± SEM (*n* = 3), *p* values: *** *p* < 0.001, * *p* < 0.05 vs. control (**upper** panel). Fluorescence imaging was done in these cells stained with DCFDA and cells with higher green fluorescence showed higher cellular ROS accumulation (lower panel). (**D**) Mitochondrial ROS/mitochondrial superoxide levels were assessed using MitoSox Red probe (5 µM for 30 min) by measuring fluorescence at 510/580 nm (excitation/emission) using a fluorescent plate reader after PC cells treated with Nic (5 μM and10 μM) for 24 and 48 h. The results were expressed as relative fluorescence intensity units. Values are expressed as mean ± SEM (*n* = 3), *p* values: *** *p* < 0.001, * *p* < 0.05 vs. control (left panel). Representative image of MitoSOX red staining of superoxide radicals in Nic treated COLO 357 cells for 24 h by confocal microscopy (right panel). (**E**) Mitochondria fragmentation was determined by using Mito-tracker Red (100 nm for 30 min) in COLO 357 and SW1990 cells treated with Nic (5 µM and 10 µM) for 24 h. (**F**) Representative Western blot images showing the expression pattern of mitochondria fission protein Drp1 at different time points (1, 3, 12, 24, and 48 h) in COLO 357, SW1990 cells treated with Nic (10 µM). Membrane was stripped and re-probed with β-actin as an internal loading control.

**Figure 3 cancers-13-03105-f003:**
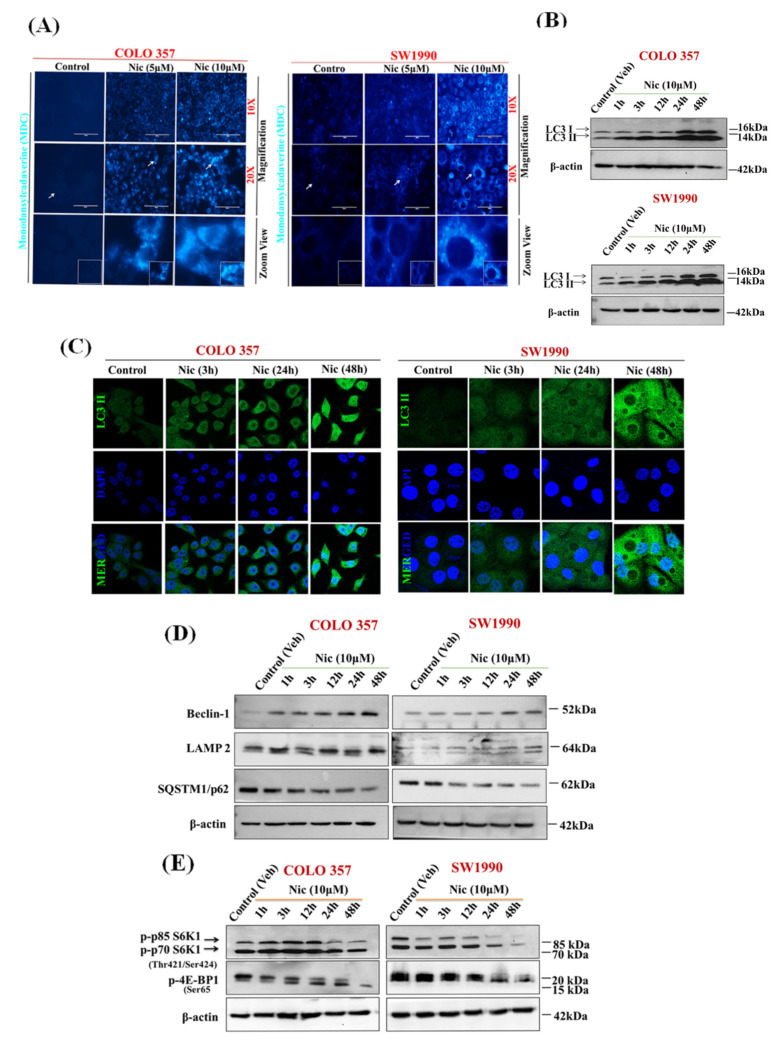
Nic stimulates autophagy through inhibition of mTORC1 signaling in PC cells. (**A**) Increased numbers of autophagic vacuoles represented by monodansylcadaverine (MDC) staining are seen with Nic (5 μM or 10 µM) by fluorescence microscope (10×; 20× magnification). (**B**) Representative Western blot images showing the expression of LC3I/II in COLO 357 and SW1990 cells treated with Nic (10 μM) for various time points, i.e., 1, 3, 12, 24, and 48 h. β-actin was used as an internal loading control. (**C**) Representative images showing the LC3 punctate were observed by confocal microscopy. (**D**,**E**) COLO 357 and SW1990 cells were treated with Nic as described previously. Representative Western blot images showing the expression of a protein involved in mTORC1 dependent autophagy such as Beclin-1, LAMP2 and P62 (**D**) p-P70S6K1 and p-EBP1 (**E**), at different time points (1, 3, 12, 24 and 48 h) in COLO 357, SW1990 cells treated with Nic (10 µM). Membrane was stripped and re-probed with β-actin to correct for loadings.

**Figure 4 cancers-13-03105-f004:**
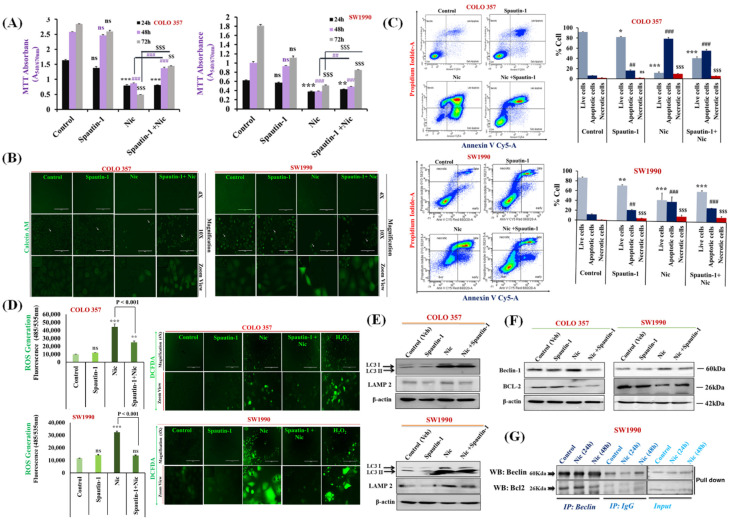
Nic-induced autophagy leads to PC cell death by disrupting Beclin1–BCL2 interaction. (**A**) Effect of functional blockage of autophagy via spautin-1 on Nic-treated PC cells resulted in growth promotion. COLO 357 and SW1990 cells were pretreated with spautin-1 for 2 h, followed by Nic (10 µM) for 24, 48, and 72 h and cell viability was assessed by MTT assay. Values are expressed as mean ± SEM (*n* = 3), *p* values: *** *p* < 0.001, ** *p* < 0.01 vs. control (24 h); *p* values: **^###^**
*p* < 0.001, **^##^**
*p* < 0.001 vs. control (48 h); *p* values: ^$$$^
*p* < 0.001, ^$$^
*p* < 0.001 vs. control (72 h). (**B**) Representative images showing the Calcein-AM-stained cells, original magnification 4×; 10×. (**C**) Protective effect of autophagy inhibition on Nic-induced apoptosis was determined by flow cytometric analysis of PC cell lines after staining with annexin-V Cy-5/PI (AV^+^/PI—intact cells; AV/PI^+^—nonviable/necrotic cells; AV^+^/PI and AV^+^/PI^+^—apoptotic cells) in different groups, i.e., spautin-1, Nic and combination of Spautin-1 and Nic (left panel). Quantitative analysis of these micrographs was shown as mean ± SEM, *p* values: *** *p* < 0.001, ** *p* < 0.01, * *p* < 0.05 vs. live control cells; **^###^**
*p* < 0.001, **^##^**
*p* < 0.001 vs. apoptotic control cells; ^$$$^
*p* < 0.001 vs. necrotic cells (right panel). (**D**) Intracellular ROS levels in cells treated with spautin-1, Nic and their combination were evaluated using DCFDA fluorogenic dye (10 µM for 30 min) by measuring fluorescence at 485/535 nm (excitation/emission) using a fluorescent plate reader (left panel). The results are expressed as relative fluorescence intensity units. Values are expressed as mean ± SEM (*n* = 3), *p* values: *** *p* < 0.001, ** *p* < 0.01 vs. control. Fluorescence imaging was performed in these cells stained with DCFDA (right panel), original magnification 4×. (**E**,**F**) Representative western blot images showing the expression of LC3I/II and LAMP2 (**E**); Beclin-1 and BCL2 (**F**) in COLO 357 and SW1990 cells treated with spautin-1, Nic and their combination. Membrane was stripped and re-probed with β-actin to correct for loading control. (**G**) Representative western blot images showing the expression of Beclin-1 and BCL2 in cells treated with Nic (10 µM; 24 and 48 h) and lysates were immunoprecipitated with Beclin-1 antibody and analyzed by western blotting for BCL2 interaction.

**Figure 5 cancers-13-03105-f005:**
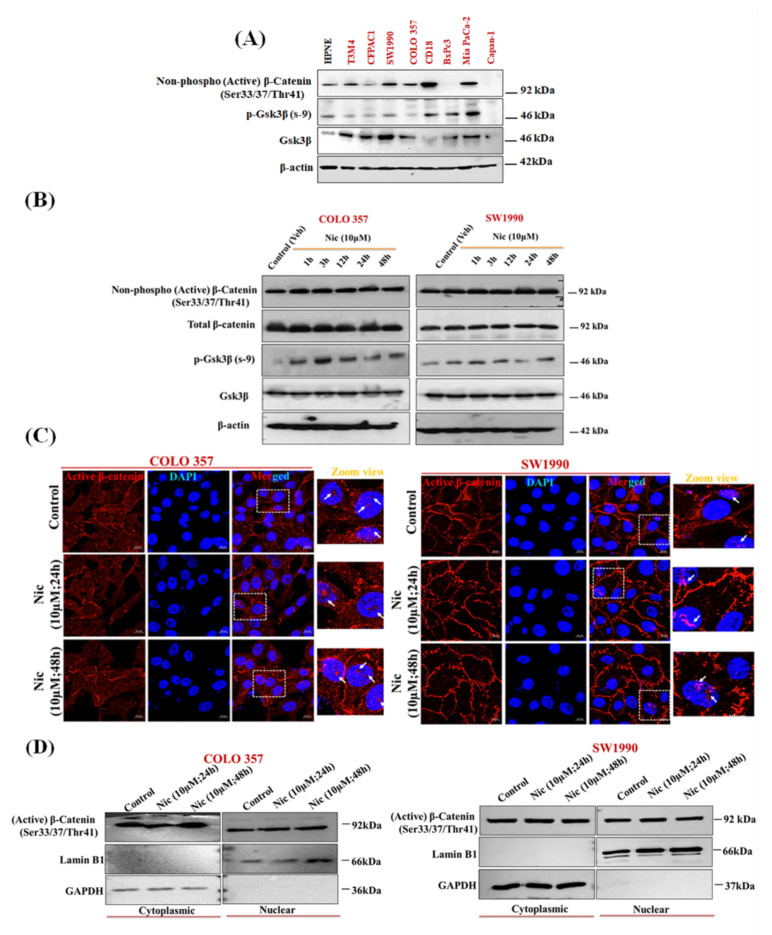
Nic promotes the inactivation of Gsk3β, and its regulation is independent of β-catenin in PC cells. (**A**,**B**) Representative Western blot images showing the expression of p-Gsk3β, Gsk3β and β-catenin (active) in different PC cell lines (**A**), p-Gsk3β, Gsk3β, p-β-catenin (active) and β-catenin (total) at different time points (1, 3, 12 24, and 48 h) in COLO 357, SW1990 cells treated with Nic (10 µM) (**B**). β-actin was used as loading as an internal loading control. (**C**) Representative confocal laser scanning microscope (CLSM) images showing nuclear translocation of p-β-catenin (active) in Nic treated COLO 357, SW1990 cells for 24 and 48 h. (**D**) Representative Western blots showing the expression of p-β-catenin (active) in cytosolic and nuclear extract of COLO 357, SW1990 cells treated with Nic (10 µM for 24 and 48 h) as compared to control. GAPDH was used as purity control for cytosolic protein extraction, whereas Lamin B1 was used as purity control for nuclear protein extraction.

**Figure 6 cancers-13-03105-f006:**
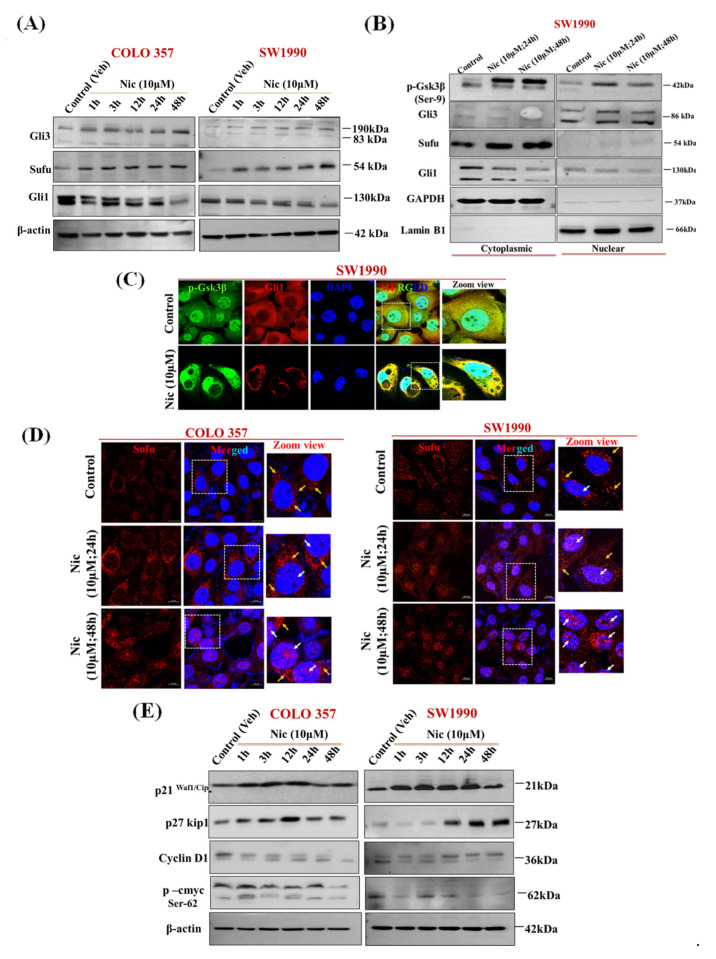
Nic inhibits Hh/Gli1 cascade via upregulation of Sufu and Gli3, a negative regulator of Hh signaling in PC cells (**A**) Representative Western blot images showing the expression of Hh signaling molecules such as Gli3 (transcription repressor), Sufu (a negative regulator of the pathway) and Gli1 (transcription activator) at different time points (1, 3, 12, 24, and 48 h) in COLO 357, Scheme 1990. cells treated with Nic (10 µM). β-actin was used as an internal loading control. (**B**) Representative Western blots showing the expression of p-Gsk3β, Sufu, Gli3, and Gli1 in cytosolic and nuclear extract of SW1990 cells treated with Nic (10 µM; 24 and 48 h) as compared to control. GAPDH was used as purity control for cytosolic protein extraction, whereas, Lamin B1 was used as purity control for nuclear protein extraction. (**C**) Representative CLSM images showing the co-expression of p-Gsk3β and Gli1 in SW1990 cells treated with Nic (10 µM) for 24 h as compared to control. (**D**) Representative CLSM images showing nuclear translocation of Sufu in COLO 357, SW1990 cells treated with Nic for 24 and 48 h. (**E**) Representative Western blot images showing the expression of a downstream target molecule of Gsk3β such as p21, p27, and cyclin D1 as well as Hh signaling downstream molecule like p-cMyc at different time points (1, 3, 12, 24, and 48 h) in COLO 357, SW1990 cells treated with Nic (10 µM). β-actin was used as an internal loading control.

**Figure 7 cancers-13-03105-f007:**
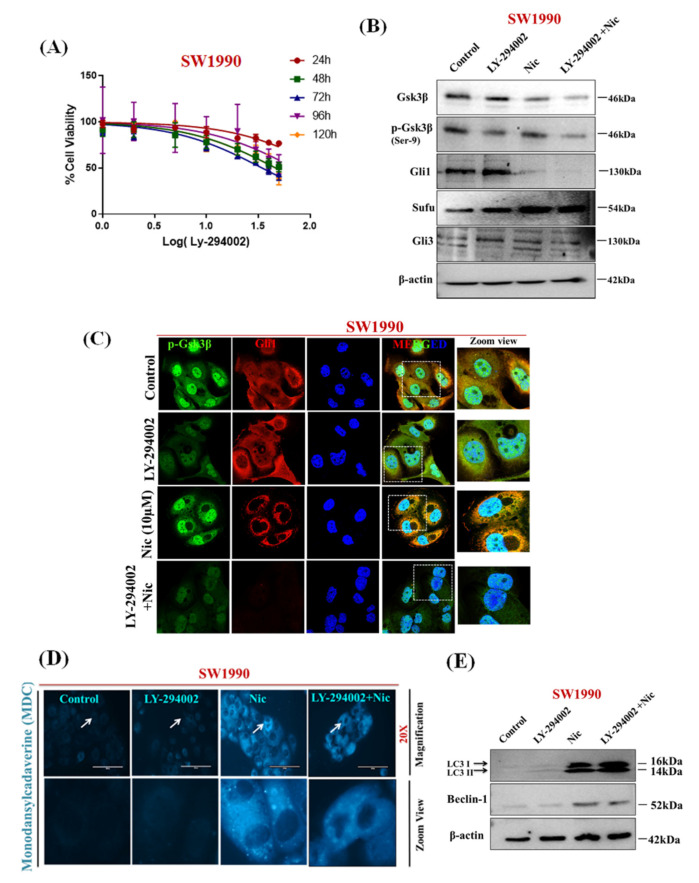
Nic induced Gsk3β inactivation inhibits Gli1 activation, negatively impacting Hh signaling and promoting autophagy-mediated apoptosis (**A**) Cell viability curve for LY-294002 (Gsk3β-activator) on SW1990 following 24, 48, 72, 96, and 120 h exposure followed by MTT assay. Values are expressed as mean ± SEM (*n* = 5). (**B**) Representative Western blot images showing the expression of p-Gsk3β, Gsk3β, Gli1, Sufu, and Gli3 in SW1990 cells treated with vehicle or LY-294002 (30 µM) or Nic (30 µM) alone or in combination with LY-294002 for 24 h. (**C**) Representative CLSM images showing the co-expression of p-Gsk3β and Gli1 in SW1990 cells treated with vehicle or LY-294002 (30 µM) or Nic (10 µM) alone or with LY-294002 for 24 h. (**D**) Autophagic vacuoles represented by MDC staining are seen in different groups as mentioned in the figure by fluorescence microscope (20× magnification) (**E**) Representative Western blot images showing the expression of autophagy markers i.e., LC3I/II and Beclin-1. β-actin is used as an internal loading control.

**Figure 8 cancers-13-03105-f008:**
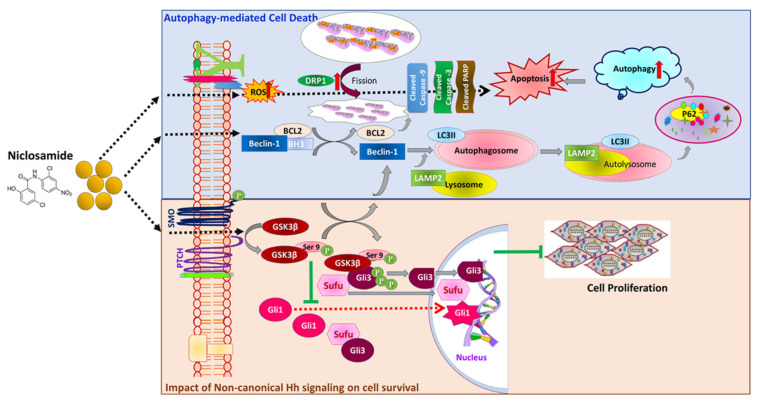
Graphical representation of overall niclosamide-mediated PC cell death mechanism(s) in pancreatic cancer: Our data demonstrated the potent anti-cancer effect of Nic in PC cells by inactivation of Gsk3β by inhibitory phosphorylation of its Ser-9. Up regulation of p-Gsk3β potentially modulates molecular signaling(s), including Hh/Gli cascade and mTORC1-dependent autophagy. Impact on non-canonical Hh signaling, the upregulation of Sufu, and nuclear accumulation of Gli3 with concurrent downregulation of Gli1 that inhibit Hh signaling and ultimately led to inhibition of PC cell proliferation/survival. Nic-induced cell death was associated with the involvement of mitochondrial stress-mediated apoptosis and mTORC1-dependent autophagy by inhibiting the interaction between Beclin-1 and BCl2 provides the cross regulatory mechanism of autophagy and apoptosis.

## Data Availability

The data presented in this study are available in “Repurposing Niclosamide for Targeting Pancreatic Cancer by Inhibiting Hh/Gli Non-Canonical Axis of Gsk3β” or in the supplementary of the same article.

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
