# Peer review of "Repurposing Niclosamide for Targeting Pancreatic Cancer by Inhibiting Hh/Gli Non-Canonical Axis of Gsk3β"

_cancers, 2021, doi:10.3390/cancers13133105_

Round 1

Reviewer 1 Report

The authors in the current manuscript have detailed a study identifying the previously reported anti cancer properties of Niclosamide. The drug has been shown to have anticancer properties in other cancers, however its role in PDAC has not been determined. 

Minor Points: 

  1. HPNE cells are predominantly representative of the epithelial and endocrine portion of the pancreas. A better and more clinically relevant control for this study would be the use of Human Pancreatic Ductal cell line. Have the authors considered the effect of Nic on the normal ductal cells?
  2. Could the authors comment on the degree of differentiate of these two cell lines? Aggressive PDAC models usually consist of cells that are dedifferentiatied thereby increasing their ability to metastasize and invade blood vessels? Does Nic have a better effect on de differentiated cells as compared to well differentiated cells?

Major Points:

  1. The authors need to demonstrate the effectiveness of the drug in an in vivo model of the disease. Without effectively demonstrating that the drug in a clinically relevant mouse model of the disease.
  2. The authors have not demonstrated the effectiveness of this drug on the tumor microenvironment in PDAC. Since fibroblasts occupy almost 90% of the tour architecture at the time when the disease is detected it is imperative that the authors show some studies that ascertain to the role of the drug on the tumor microenvironment. Perhaps a significant study would be to determine if the drug also affects the stellate cell population in the TME? Does it decrease its viability or change its activation state?

Author Response

The authors in the current manuscript have detailed a study identifying the previously reported anti cancer properties of Niclosamide. The drug has been shown to have anticancer properties in other cancers, however its role in PDAC has not been determined. 

Minor Points: 

Query1: HPNE cells are predominantly representative of the epithelial and endocrine portion of the pancreas. A better and more clinically relevant control for this study would be the use of Human Pancreatic Ductal cell line. Have the authors considered the effect of Nic on the normal ductal cells?

Reply: We thank reviewer for this important and relevant concern. In agreement with reviewer’s concern, along with HPNE, we have used human pancreatic ductal cells (HPDE), a normal pancreatic ductal cell line, to demonstrate the effect of Nic on cell viability.  We observed that the IC50 of Nic for HPDE cells was 155.3 µM for 24h, 60.86 µM for 48h, and 49.18 µM for 72h respectively, which was significantly higher than IC50 value of Nic for pancreatic cancer cells, i.e., ~10µM. Therefore, our results demonstrated that Nic has an anti-proliferative effect specifically on PC cells without affecting the normal pancreatic cells. As per reviewer's suggestion, this data has been incorporated in Figure 1B and mentioned in the “material and methods” (page no. 3; line no. 116) and “results section” (page no. 7; line no. 303-304) in the revised manuscript.

Query 2: Could the authors comment on the degree of differentiate of these two cell lines? Aggressive PDAC models usually consist of cells that are dedifferentiated thereby increasing their ability to metastasize and invade blood vessels? Does Nic have a better effect on de differentiated cells as compared to well differentiated cells?

Reply: We again appreciate the reviewer for the highly relevant and insightful comment. Among the pancreatic cancer cell lines used in this study, COLO357 is a well-differentiated cell line whereas, SW1990 is a moderately differentiated cell line. We have incorporated this information in  the revised manuscript (“material and methods” section; page no. 3; line no. 126-128). In agreement with the reviewer's opinion, we have also performed the cell viability assay to determine the effect of Nic on two poorly differentiated cell lines, PANC-1 and MIA PaCa-2, and one more well-differentiated cell line Capan-1 at different Nic concentrations (5µM, 10µM, 15µM, and 20 µM) (Sup Fig. 1). Likewise, significant suppression in growth was observed in these cell lines after 24h treatment, and this effect was more prominent after 48h treatment. Further, flow cytometry analysis on poorly differentiated PANC-1 and AsPC-1 cell lines suggested that Nic-mediated growth inhibitory effect is associated with apoptosis induction (Sup Fig. 6 A, B). For clarity, this data has been incorporated in the revised supplementary information (Supplementary Figure 1 & Supplementary Figure 6 A, B) and revised manuscript (“material and methods” section, page no. 3; line no. 115, 130-132; “results” section, page no. 7; line no. 306-310; page no. 8; line no. 376-380).

  1. Zhang SN, Huang FT, Huang YJ, Zhong W, Yu Z. Characterization of a cancer stem cell-like side population derived from human pancreatic adenocarcinoma cells. Tumori Journal. 2010 Nov;96(6):985-92.
  2. Sipos B, Möser S, Kalthoff H, Török V, Löhr M, Klöppel G. A comprehensive characterization of pancreatic ductal carcinoma cell lines: towards the establishment of an in vitro research platform. Virchows Archiv. 2003 May;442(5):444-52.s
  3. Zinn R, Otterbein H, Lehnert H, Ungefroren H. RAC1B: A guardian of the epithelial phenotype and protector against epithelial-mesenchymal transition. Cells. 2019 Dec;8(12):1569.

Major Points:

Query 1:The authors need to demonstrate the effectiveness of the drug in an in vivo model of the disease. Without effectively demonstrating that the drug in a clinically relevant mouse model of the disease

Reply 1: We appreciate the reviewer for the suggestion related to clinical relevance of the drug. In the present study, our major aim was to establish the efficacy of Nic as a potential candidate for pancreatic cancer therapy and to understand its detailed mechanism of action. Therefore, in this study, most of the work is performed in vitro on the cell line models.  Niclosamide delivery, either oral or systemic, is challenging due to its insoluble nature, leading to its reduced oral bioavailability that limits it’s in vivo efficacy [1-3].  However, we are working on synthesizing polyanhydride-coated nano formulated Nic, and in future studies, we will perform its efficacy in vivo-PC models and will altogether be a detailed study and a part of separate manuscript. This study provides valuable preliminary data that makes the basis for an in-depth study of Nic in vivo in various PC animal models alone and in combination with other chemotherapeutic drugs for our ongoing research (metastatic pancreatic cancer).

  1. Zhang X, Zhang Y, Zhang T, Zhang J, Wu B. Significantly enhanced bioavailability of niclosamide through submicron lipid emulsions with or without PEG-lipid: a comparative study. Journal of microencapsulation. 2015 Jul 4;32(5):496-502.
  2. Rehman MU, Khan MA, Khan WS, Shafique M, Khan M. Fabrication of Niclosamide loaded solid lipid nanoparticles: in vitro characterization and comparative in vivo evaluation. Artificial cells, nanomedicine, and biotechnology. 2018 Nov 17;46(8):1926-34.
  3. Fan X, Li H, Ding X, Zhang QY. Contributions of hepatic and intestinal metabolism to the disposition of niclosamide, a repurposed drug with poor bioavailability. Drug Metabolism and Disposition. 2019 Jul 1;47(7):756-63.

Query 2: The authors have not demonstrated the effectiveness of this drug on the tumor microenvironment in PDAC. Since fibroblasts occupy almost 90% of the tour architecture at the time when the disease is detected it is imperative that the authors show some studies that ascertain to the role of the drug on the tumor microenvironment. Perhaps a significant study would be to determine if the drug also affects the stellate cell population in the TME? Does it decrease its viability or change its activation state?

Reply 2: We appreciate reviewer’s valid concern to determine niclosamide therapeutic efficacy on the tumor microenvironment, specifically on cancer associated fibroblasts.  In agreement, in the revised manuscript, we demonstrated the efficacy of Nic in human pancreatic cancer-associated fibroblast (CAFs) (09-11,09-17,10-32,10-15,10-03 CAFs) and normal fibroblast (09-26N) cell lines by viability assay (Sup Fig. 3). Like PC cells, we also observed a significant reduction in the viability of CAFs compared to the normal fibroblasts after 24h of Nic treatment (5µM and 10µM), and this effect was further enhanced after 48h of incubation. Notably, similar Nic concentrations were ineffective in inhibiting the growth of normal fibroblasts after 24h treatment, however, after 48h, the inhibitory effect was evident (Sup. Fig. 3A). Further, flow cytometry analysis on 10-32 CAFs cell lines suggested that Nic-mediated growth inhibitory effect is associated with apoptosis (Sup. Fig. 6C). We also investigated the effect of Nic on fibroblast activation. We observed  reduction in Alpha smooth muscle actin (α-SMA), a fibroblast activation marker, upon Nic treatment however we didn’t observed any increase or decrease in the expression of fibroblast activation marker (FAPα) (Sup. Fig. 3B). Together, this suggested that Nic treatment reduces the viability of CAFs without altering their activation status. We have provided these data in the revised supplementary information (Supplementary Figure 3 & Supplementary Figure 6C) and revised manuscript (“material and methods” section, page no. 3; line no. 122-125; “results” section, page no. 8; line no. 344-353; page no. 9; line no. 378).

Reviewer 2 Report

  1. In Figure 2, full-length PARP was not observed in control samples, and was not reduced in the Nic-treated samples, showing the discrepancies between WB and FACS analysis. These data cannot support the hypothesis authors mentioned in this manuscript (Figure 8).
  2. Authors should describe the methods of WB in the materials and methods section.
  3. Authors should show the activation of caspase 3 by WB.

Author Response

Query 1: In Figure 2, full-length PARP was not observed in control samples, and was not reduced in the Nic-treated samples, showing the discrepancies between WB and FACS analysis. These data cannot support the hypothesis authors mentioned in this manuscript (Figure 8).

Reply 1: We thank and appreciate the reviewer for the critical analysis of the experiment and the concern raised. We completely agree that there is less full-length PARP expression in the control samples compared to the treated ones. The best explanation we think of right now is that Nic treatment is leading to the overall PARP expression along with its cleavage, which might be the reason for increased apoptotic signal in the treated samples. Furthermore, for clear interpretation of the expression, we have analyzed the WB representative images with densitometry quantitation, and significant expression of full-length PARP with increased expression of cleaved PARP was noticed. This data has been incorporated in the revised supplementary information (Sup. Fig 4) and mentioned in the revised manuscript (“results” section, page no. 8, line no. 372-374).

Query 2: Authors should describe the methods of WB in the materials and methods section.

Reply 2: As per the reviewer's suggestion, we have incorporated the detailed description of the western botting experiment in the revised manuscript (“material and methods” section, page no. 4; line no. 168-192).

Query 3: Authors should show the activation of caspase 3 by WB.

Reply 3: We would like to thank the reviewer for the suggestion. In order to analyze the effect of Nic in inducing Caspase 3 activation, we performed western blotting analysis to check the expression of cleaved caspase-3 after Nic treatment (10µM; 24h, 48h) on pancreatic cancer cells (COLO357, SW1990). We observed a significant increase in caspase 3 activation as evident by cleaved caspase expression at the given concentration and timepoints. For convenience, we have incorporated these results in Supplementary Fig. 5 and described in results section, page no. 8; line no. 375-376).

Round 2

Reviewer 1 Report

The authors have satisfactorily addressed the reviewers comments

Author Response

Thank you for your valuable suggestions. We have revised our manuscript carefully.

Reviewer 2 Report

Authors' answers was not appropriate.

Author Response

Thank you for your valuable suggestions. In our revised manuscript, we provided the statistical analysis of supplementary figures 4 and 7.